# Novel measurement tool and model for aberrant urinary stream in 3D printed urethras derived from human tissue

Andrew J. Cohen[1], German Patino[2], Mehran Mirramezani[3],
Sudarshan Srirangapatanam [4], Anas Tresh[4], Bhagat Cheema[4], Jenny Tai[5],
Dylan Romero[5], Anthony Enriquez [4], Laurence S. Baskin[4], Shawn C. Shadden[3],
Benjamin N. Breyer[4,6] *

1 Brady Urological Institute, Johns Hopkins Bayview Medical Center, Baltimore, MD, United States of America, 2 Hospital Universitario San Ignacio, Bogota, Colombia, United States of America, 3 University of California, Berkeley, Department of Mechanical Engineering, Berkeley, CA, United States of America, 4 Department of Urology, University of California San Francisco, San Francisco, CA, United States of America, 5 Makers Lab Library, University of California San Francisco, San Francisco, CA, United States of America, 6 Department of Biostatistics and Epidemiology, University of California, San Francisco, CA, United States of America

* Benjamin.Breyer@ucsf.edu

**Data Availability Statement:** All relevant data are within the paper and its Supporting Information files.

## Abstract

### Background

An estimated 10% of male adults have split or dribbled stream leading to poor hygiene, embarrassment, and inconvenience. There is no current metric that measures male stream deviation.

### Objective

To develop a novel method to measure spray in normal and abnormal anatomical conformations.

### Design, setting, and participants

We developed a novel platform to reliably describe spray. We used cadaveric tissues and 3D Printed models to study the impact of meatal shape on the urinary stream. Cadaveric penile tissue and 3D printed models were affixed to a fluid pump and used to simulate micturition. Dye captured on fabric allowed for spray detection.

### Outcome measurements and statistical analysis

Spray pattern area, deviation from normal location, and flowrates were recorded. Computational fluid dynamic models were created to study fluid vorticity.

### Results and limitations

Obstructions at the penile tip worsened spray dynamics and reduced flow. Ventral meatotomy improved flowrate (p<0.05) and reduced spray (p<0.05) compared to tips obstructed

**Funding:** Specific funding for this work came as a private gift from Anita and Kevan Del Grande, awarded to BB (gift project #7028070). The funders had no role in study design, data collection and analysis, decision to publish, or preparation of the manuscript.

**Competing interests:** The authors have declared that no competing interests exist.

ventrally, dorsally or in the fossa navicularis. 3D models do not fully reproduce parameters of their parent cadaver material. The average flowrate from 3D model was 10ml/sec less than that of the penis from which it was derived (p = 0.03). Nonetheless, as in cadavers, increasing obstruction in 3D models leads to the same pattern of reduced flowrate and worse spray. Dynamic modeling revealed increasing distal obstruction was correlated to higher relative vorticity observed at the urethral tip.

## Conclusions

We developed a robust method to measure urine spray in a research setting. Dynamic 3D printed models hold promise as a methodology to study common pathologies in the urethra and corrective surgeries on the urine stream that would not be feasible in patients. These novel methods require further validation, but offer promise as a research and clinical tool.

## Introduction

Urination is an activity of daily living that greatly impacts quality of life. Even moderate levels of lower urinary tract symptoms, such as weak or split stream, are associated with anxiety and depression [1]. An estimated 10% of adult males have split stream or dribble [2]. Men with stream alterations are more likely to sit to urinate, and report degraded urinary quality of life. Stream deviation is more likely for patients with a history of urethral reconstruction [3]. While uroflowmetry provides information regarding the functioning of the bladder and outlet; it provides no data about the shape, morphology, or dispersion of the urinary stream [4]. Historically, urinary drop spectrometry assessed the amplitude of urinary drop patterns [5,6] but there is no contemporary instrument that measures the degree of stream variation after exiting the penis, nor is there a definition of normal in this context.

The effect of anatomic variation on stream characteristics is understudied. There is wide variation in the shape and size of the meatus in the general population with increasing attention to classifying meatal conformations and associated pathology [7]. With the assumption of the urethral outlet as an elliptical orifice, computational modelling has simulated a wavelike shape of the male urinary stream [8]. However, there is a paucity of work on variant anatomy of the meatus and in particular the effect, if any, on urinary stream dynamics.

Pathologic meatal shapes may lead to undesirable stream deviation. The risk of meatal stenosis is estimated to be 0.66% in circumcised men [9]. Meatal stenosis leads to weak stream and voiding dysfunction if left untreated [10]. Meatoplasty is the treatment of choice for strictures at the meatus, but treatment itself may lead to irregular stream [11]. Surgeries that alter the shape of the fossa navicularis or meatus may improve urine flow parameters; yet the ideal conformation to reduce spraying is unknown. While surgeons strive to create a cosmetically appealing result with unobstructed flow, the post-operative presence of spraying can unpredictable and extremely bothersome to patients. This is particularly germane to urethroplasty and hypospadias outcomes [2,3]. An obstacle to studying these issues includes: the lack of a high-fidelity measurement system, unknown definition of normal spray deviation, and ethical issues surrounding experimental changes to the urethral meatus in humans.

Our objective was to 1) develop a reliable system to measure urinary spray 2) define and measure average urine spray parameters 3) evaluate the effect of common pathologies in the distal urethra and their corrective surgeries on the urine stream 4) understand if 3D models could serve as a reliable proxy for human tissue and 5) apply computational fluid dynamic

modeling to explore the flow impacts of conformational changes. Given the complexities of micturition, heterogeneity of the voiding experience [12], logistical challenges of clinically observing spray, and inability to experimentally alter meatal shapes in humans we elected to use cadaveric and 3D printed urethral models in this pilot study.

## Methods

### Summary

Four cadaver specimens were employed for baseline flow measurements, and then conformational tip changes were made to understand resultant effects on flowrates and spray. In parallel, a 3D printed model of each cadaveric urethra was run through the same experiments. Conformational change of the model was limited to a reduction in aperture of 40%. We simultaneously molded urine flow *in-silico* using computational fluid dynamics.

### Cadaveric tissues

Fresh cadaveric tissues were provided from mid-bulb to glans on block. Minimal clinical details of donors were provided, but none had documented penile abnormalities or prior urethral surgery. UCSF's Committee on Human Research has deemed that our research project does not require IRB review based on two points: 1) Our research is not regulated by the FDA and 2) We received de-identified data or specimens. The committee on University of California Anatomical Materials Programs Standards and Guidelines approved the use of anatomical material.

### Flow replication experiments

We utilized a MedAmicus 4114UF Lumax Cystometry System (Enpath Medical, Minnesota) for uroflowmetry measurements. We employed a DEP-4000 DC Water Pump with Controller (Uniclife, Colorado) to generate controlled flows through 18 Fr silicone tube affixed to a cadaveric or 3D printed model. (Fig 1). Four varied flow scenarios were calibrated with target maximum flow rates of 7, 16, 22 and 30 ml/sec with varied flow patterns, respectively (S1 Appendix). These were selected to mirror flow patterns seen clinically. The flow rate target of 22 mL/sec was considered the 'normal scenario.' These flow scenarios were controlled by adjusting the digital pump via a timer, replicating each scenario during all experiments. In other words, the 'bladder function' was standardized for each scenario.

Each cadaveric specimen was affixed to the fluid pump and 3 identical experiments to test reproducibility were performed for each of the four flow scenarios. Uroflowmetry was recorded and spray patterns simultaneously captured by photo and video (Nikon d7500, New York). All four cadaveric specimens were used.

### Spray patterns

A 62 cm by 45 cm catchment area was created using a modified light-weight plastic container found to be compatible with our uroflowmetry scale. Easy Trap Duster Cloth (3M, Minnesota) sheets were placed on top of this area and used to allow for the visual capture of spray. The cloth was pre-soaked with water to reduce secondary splatter that resulted from the impact of urine on the testing apparatus. We placed the top of our detector 16.5 cm below the level of the penis to replicate a typical standard urinal height. Blue dyed water was used for flow experiments. (Fig 1) Urine flows were produced via pump as described above and video and still images captured of each result (S1 File). A measuring tape was affixed to the cloth sheets to allow for calibration between scenarios.

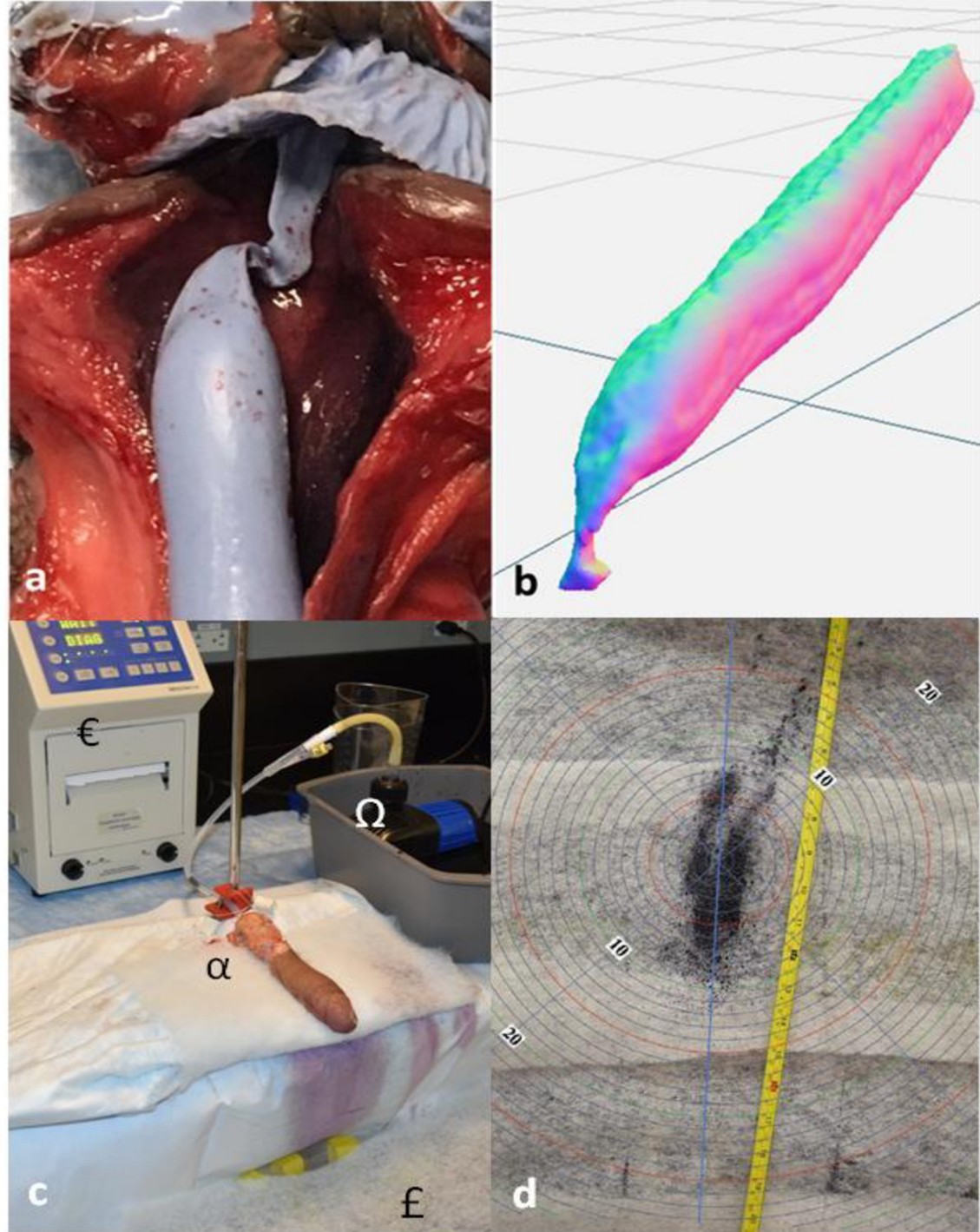

**Fig 1.** a: Silicone model of cadaveric urethra *in-situ* b: Scanned urethral lumen for computational modeling and 3D printing c: Experimental set up for flow experiments; α: Modified 18 Fr silicone tube affixed to cadaveric penis, €: MedAmicus 4114UF Lumax Cystometry System, Ω: Uniclife DEP-4000 Controllable DC Water Pump, £: Urine spray detector apparatus d: Example urinary spray pattern result of cadaveric model.

In post-experiment processing, concentric measuring rings were digitally applied. We defined a target, or the area of maximal urine saturation, as coordinates 0,0. This location was derived from images collected during initial cadaveric specimen testing, which was not statistically different among the 4 specimens. For analysis purposes, all subsequent experiments were defined relative to this location. For ease of analysis we also defined the linear displacement (as the absolute value of x deviation + absolute value of y deviation). Furthermore, we defined the spray area as the rectangle in which dye was seen immediately after experimentation. For presenting graphical data, the area of spray was normalized to the average value of the spray area from the normal scenario using penile specimens 1–4 (38.6 cm$^2$).

## 3D models

Silicone Tin-Cure Rubber with shore 30A hardness (Smooth-On, Inc., Pennsylvania) was used to cast the lumen of each cadaveric urethra. (Fig 1). A small proximal urethrotomy was used to remove the soft, silicone cast and subsequently repaired with 4–0 vicryl suture. This area was always proximal to the insertion of our 18 Fr tubing. The silicone molds of urethral were digitized and converted to a 3D printable model (Matter and Form scanners, Toronto). The 3D models were prepared and smoothed using post-processing software (Meshmixer, Autodesk, California).

The 3D model of the urethra was subtracted from a cylindrical shape. The resulting effect left a solid cylinder with a hollow interior cavity in the shape of the original urethra model. After verifying size and fit, a proximal extension piece was designed to aid in the insertion of 18 Fr tubing for flow experiments. The models were printed on the 3D printer (Lulzbot Mini, Colorado) using polylactic acid material (PLA). A 3D model of each of the 4 cadaveric specimens was created.

## Meatal alterations

Penis specimen 1 was sacrificed as a test bed for troubleshooting silicone mold removal techniques; hence, we performed a series of experiments to alter the natural shape of the 3 remaining cadaveric specimens. Conformation changes of the distal urethra were performed surgically as described in (Fig 2) in a sequential fashion. Ultimately the following scenarios was applied: 1. a distal fossa extrinsic obstruction of 14 French (Fr), 2. a distal fossa extrinsic obstruction of 14 Fr and meatus 12 Fr dorsal occlusion, 3. a distal fossa extrinsic obstruction of 14 Fr and meatus 12 Fr ventral occlusion, 4. a 12 Fr dorsal meatus occlusion, 5. a 8 Fr dorsal meatus occlusion, 6. a 12 Fr ventral meatus occlusion, 7. a 8 Fr ventral meatus occlusion, and 8. a formal meatotomy with urethral advancement using 4, 4–0 vicryl sutures.

Extrinsic compression was applied using a ring clamp. The most proximal portion of the ring clamp (1cm in length) was placed approximately 1.5 cm from the urethral meatus, at the position of the fossa navicularis. The calibrating catheter ensured the occlusion was consistent across experiments. A 14 Fr occlusion was fashioned so the ring clamp was tightened until a 14 Fr catheter would pass with resistance but a 16 Fr failed to pass Meatus occlusions were performed at the tip of the meatus using running 4–0 vicryl suture with a calibrating catheter in place. In this case, an 8 Fr meatus was tailored such as a 8 Fr catheter would pass with resistance but a 10 Fr failed to pass. Such sutures were limited to the distal 1 cm of urethra and placed mucosa-to-mucosa. No efforts were made to purposefully alter the natural shape of the urethra, just its size The experiments were performed in sequence from least to increasingly invasive with meatoplasty representing the final step. For each conformational change, the specimens were affixed to the fluid pump and run through the 4 flow scenarios (S1 Appendix) for testing of flowrates, spray area, and distance from target.

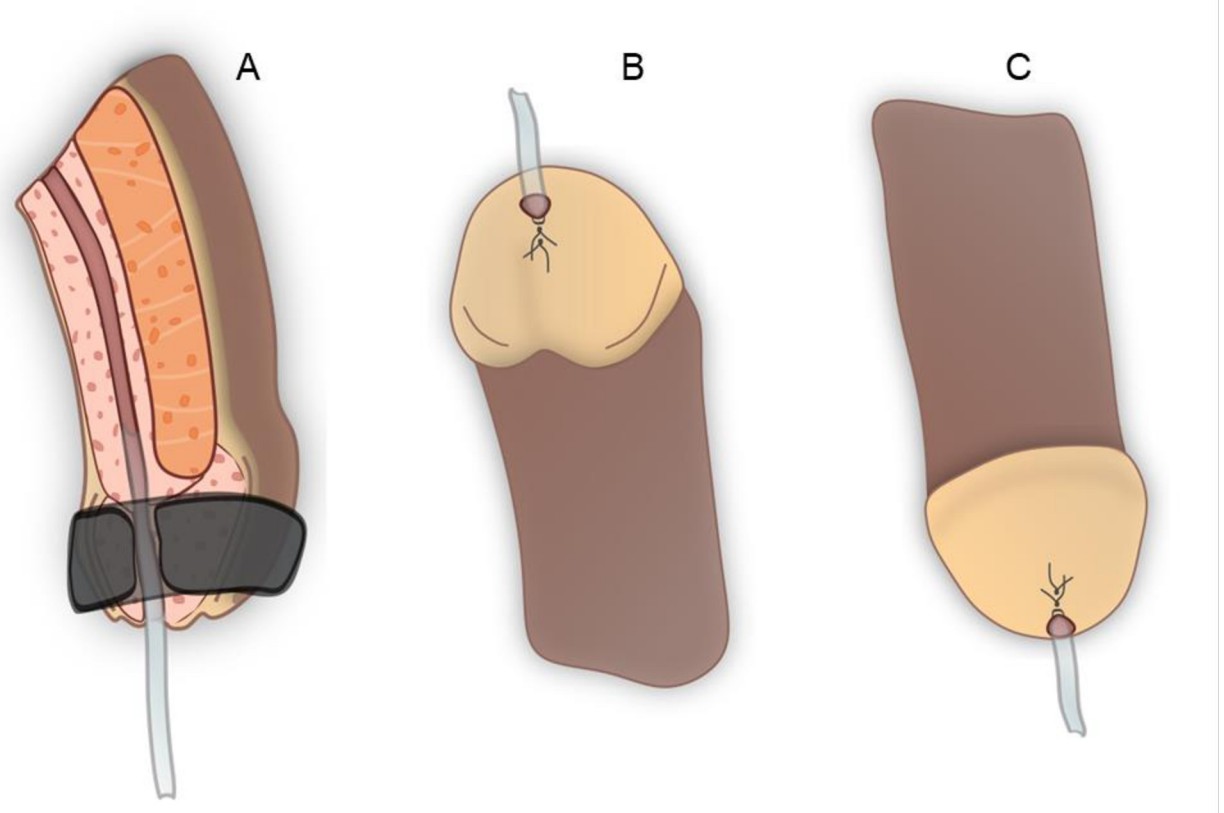

**Fig 2. Conformational changes performed to reduce lumen size.** A: Extrinsic compression applied to penile tissue using ring clamp with calibration over catheter, B: Creation of ventral narrowing at meatal tip with calibrating catheter in place, C: Creation of dorsal narrowing at meatal tip with calibrating catheter in place.

The only confirmation change created for the 3D printed models was a reduction in aperture of the tip by 40%. These were printed on the 3D printer (Lulzbot Mini, Colorado) using polylactic acid material (PLA) using the same technique as previously described.

## Computational fluid dynamic models

Using the digital urethral model from specimen 2, a marker of turbulence and pressure was computed at the approximate level of the fossa navicularis. Methodology used to model urine flows has been previously described [13]. Simvascular was utilized for simulations [14]. Streamlines and corresponding vorticity patterns were computed to demonstrate local flow disturbances induced from the conformational change. Specifically, vorticity is a measure of the instantaneous rotation of a fluid parcel. A 1 cm long stricture was modeled at the distal urethral location, mirroring our experiments in cadaveric tissue. Computational experiments were repeated with increasingly obstructed models to assess changes in pressure and vorticity.

## Analysis

Descriptive statistics were used to analyze results with significance level set at $p<0.05$. Nonparametric Kruskal–Wallis testing was used to compare flow-rates and spray area among groups. Stata 13.1 (College Station,Texas) was used for all calculations.

## Results

### Flow replication experiments

Cadaver donors were on average 69.5 years old (IQR 60–77). Meatus width was 9mm (IQR: 8–10.5) and height 9.5mm (IQR: 8.5–11.5). Available urethral length was on average 15.5 cm (IQR 12.7–17.5). The average flow rate and spray area for the normal scenario among penile cadaveric specimens was 22.2 ml/sec and 38.6 cm$^2$, respectively (Table 1).

### 3D models

3D printed models had significantly lower flow-rates (11.6 ml/sec lower on average; *p<0.01*) than their cadaveric counterparts (Table 1). The reduction in flow rates for models vs. matched cadaveric tissues held for all flow scenarios (S1 Appendix). For models 1, 3 & 4 the spray area of the model was greater than that of the cadaveric tissues during the normal scenario (all *p = 0.03 or less*). Indeed, in 68.8% of experiments, the spray area of the 3D model was greater than that of the native urethra.

### Meatal alterations

Linear displacement of the focal point of urine was observed to differ by scenario and meatal conformation. For any altered anatomy, the median displacement was 15 cm (IQR: 9.5–32). A decrease in flow led to a decrease in observed linear displacement; median displacement for all high pressure vs. very low pressure scenarios was 16cm (IQR: 6–40) versus 10cm (IQR: 7.5–14.5); p = 0.01, respectively. In the normal scenario, the 90th percentile displacement represented a distance 25 cm away from the target whereas for scenario 2 (low pressure) the 90th

**Table 1. Flow-rates and spray areas recorded by novel detection system in cadaveric and 3D printed models.**

|  | Specimen (#) | Cadaver Qmax (ml/s) | Matched Model Qmax s (ml/s) | P-value | Cadaver Spray Area (cm$^2$) | Matched Model Spray Area (cm$^2$) | P-value |
|---|---|---|---|---|---|---|---|
| Normal Flow Scenario | 1 | 21.7 | 19.0 | 0.04 | 38.5 | 70 | 0.02 |
| | 2 | 21.0 | 16.3 | 0.03 | 48 | 31.3 | 0.02 |
| | 3 | 23.3 | 4.6 | 0.04 | 48 | 70 | 0.03 |
| | 4 | 22.9 | 6.3 | 0.05 | 20 | 42.5 | 0.03 |
| | Average | 22.2 | 11.6 | <0.01 | 38.6 | 53.4 | 0.11 |
| | Flow scenario $^\alpha$ | | | | | | |
| Specimen 2 | 1 | 21.0 | 16.3 | 0.03 | 48 | 31.3 | 0.03 |
| | 2 | 15.3 | 11.3 | 0.04 | 44 | 60 | 0.03 |
| | 3 | 28 | 22 | 0.04 | 48 | 108 | 0.03 |
| | 4 | 7.7 | 6 | 0.03 | 18 | 24.5 | 0.03 |
| Specimen 3 | 1 | 23.3 | 4.6 | 0.05 | 48 | 70 | 0.03 |
| | 2 | 16 | 3.3 | 0.05 | 45 | 78 | 0.03 |
| | 3 | 31.3 | 5.2 | 0.05 | 60.5 | 138 | 0.03 |
| | 4 | 7 | 1.7 | 0.03 | 18 | 9 | 0.03 |
| Specimen 4 | 1 | 22.9 | 6.3 | 0.05 | 20 | 42.5 | 0.03 |
| | 2 | 15.9 | 2.4 | 0.05 | 31.5 | 16 | 0.03 |
| | 3 | 31.2 | 4.4 | 0.05 | 33 | 20 | 0.03 |
| | 4 | 7.8 | 1.6 | 0.05 | 31.5 | 10 | 0.03 |

$^\alpha$ Penis specimen 1 was sacrificed as a test bed for troubleshooting silicone mold removal techniques after baseline data collection; Qmax—Maximum urinary flowrate.

percentile displacement was 107 cm away from target. Ventral obstruction led to the most linear displacement.

Cadaveric specimens with experimentally obstructed tips experienced a 12.4 ml/sec (65.5%) reduction in flow rate across all scenarios. We observed a 73% increase in spray area for obstructed tips. Under normal flow conditions, obstructed tips on average lead to 7 x the spray area. The 8 Fr ventral meatus occlusion had the most visually apparent impact.

Meatoplasty maintained or increased flow rates compared to unaltered tissue (Table 2). In 75% of experiments, meatoplasty increased resultant spray area compared to unaltered tips. For all normal or high-pressure scenarios, meatoplasty increased spray in 100% of cases. When compared to tips obstructed ventrally, dorsally or in the fossa naviculars, meatoplasty reduced spray (p<0.05). Meatoplasty did not significantly affect linear displacement (p = 0.71) compared to cadaveric tissues in the normal scenario. Fig 3 visually summarizes the experimental findings from penis 2 during flow scenario 1. The flow rates of 3D printed models with a 40% reduction in aperture size, resulted in an average decrease in flow of 6.7 ml/sec across all flow scenarios (68.5% reduction).

## Computational fluid dynamic models

The urethral model was discretized into a finite element mesh and simulation proceeded with conditions under the normal flow scenario. Dynamic modeling found with increasing distal obstruction, higher relative vorticity was observed at the urethral tip. Intraurethral pressure also increased with worse obstruction. (Fig 4).

## Discussion

We present a number of novel methodologies and applications of technology to better understand urinary dynamics. To our knowledge, our spray detector is the first device created to quantify urine spray. Even unaltered cadaveric specimens mirroring normal clinically derived flowrates lead to an average spray area of 38 cm$^2$. Increasing ventral urethral occlusion leads to linear displacement and increased spray. Meatotomy maintained or improved flowrates with little linear displacement but increased spray compared to unaltered specimens. 3D models did not exactly replicate the findings in cadaveric tissues; however increasing obstruction of

**Table 2. Impact of meatoplasty on flowrates and spray area in a cadaveric model.**

|  | Flow scenario | Cadaver Qmax (ml/s) | Meatoplasty Qmax (ml/s) | P-value | Cadaver Spray Area (ml/s) | Meatoplasty Spray Area (ml/s) | P-value |
|---|---|---|---|---|---|---|---|
|  | 1 | 21 | 26 | <0.01 | 48 | 78 | 0.03 |
| Specimen 2 | 2 | 15.3 | 16.3 | 0.09 | 44 | 60 | 0.03 |
|  | 3 | 28 | 33.3 | 0.05 | 48 | 71.5 | 0.03 |
|  | 4 | 7.7 | 7 | 0.11 | 18 | 15 | 0.04 |
|  | 1 | 23.3 | 25.7 | 0.04 | 48 | 49.5 | 0.03 |
| Specimen 3 | 2 | 16 | 16 | 0.82 | 45 | 8 | 0.03 |
|  | 3 | 31.3 | 30.7 | 0.50 | 60.5 | 96 | 0.03 |
|  | 4 | 7 | 4.3 | 0.11 | 18 | 30 | 0.03 |
|  | 1 | 22.9 | 23.8 | 0.28 | 20 | 25 | 0.03 |
| Specimen 4 | 2 | 15.9 | 15.5 | 0.18 | 31.5 | 36 | <0.01 |
|  | 3 | 31.2 | 31.5 | 0.51 | 33 | 40 | <0.01 |
|  | 4 | 7.8 | 9.4 | 0.05 | 31.5 | 20 | <0.01 |

Qmax—Maximum urinary flowrate.

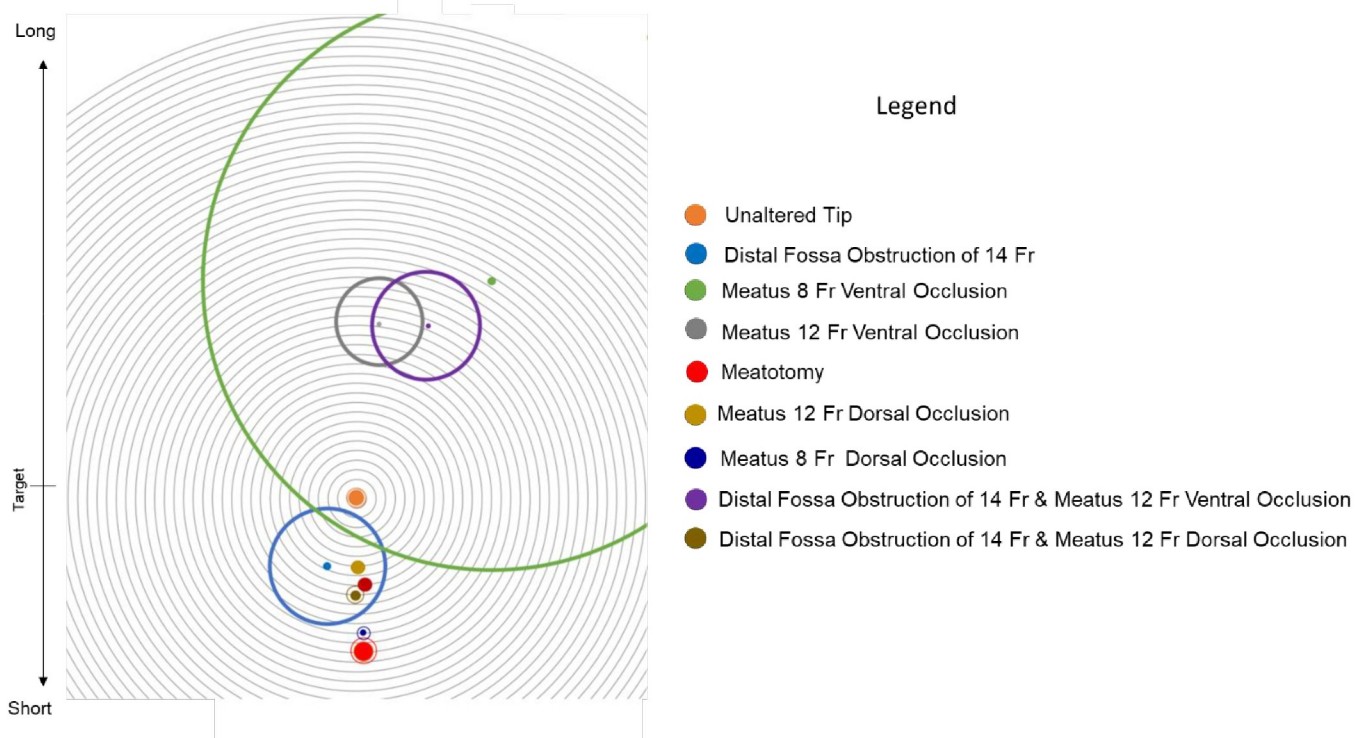

Fig 3. a) Graphical Representation of Voiding Strength and Spray Stratified by Tip Alternations for Specimen 2. b) Relative Voiding Characteristics Specimen 2.

| Tip Alteration | Relative Flow Rate, % | Relative Spray, % | Relative Distance*, cm |
|---|---|---|---|
| Unaltered | Ref | Ref | Ref |
| Distal Fossa Obstruction of 14 Fr | 52 | 748 | -11 |
| Meatus 8 Fr Ventral Occlusion | 6 | 3740 | 38 |
| Meatus 12 Fr Ventral Occlusion | 28 | 571 | 23 |
| Meatotomy | 120 | 203 | -18 |
| Distal Fossa Obstruction of 14 Fr & Meatus 12 Fr Ventral | 29 | 706 | 27 |
| Distal Fossa Obstruction of 14 Fr & Meatus 12 Fr Dorsal | 60 | 117 | -10 |
| Meatus 12 Fr Dorsal Occlusion | 72 | 29 | -11 |
| Meatus 8 Fr Dorsal Occlusion | 48 | 82 | -15 |

\* Negative Distances assigned for y axis (short/long)

models corresponded to reduced flow rates. In dynamic simulation, increasing obstruction increased vorticity and intraurethral pressures.

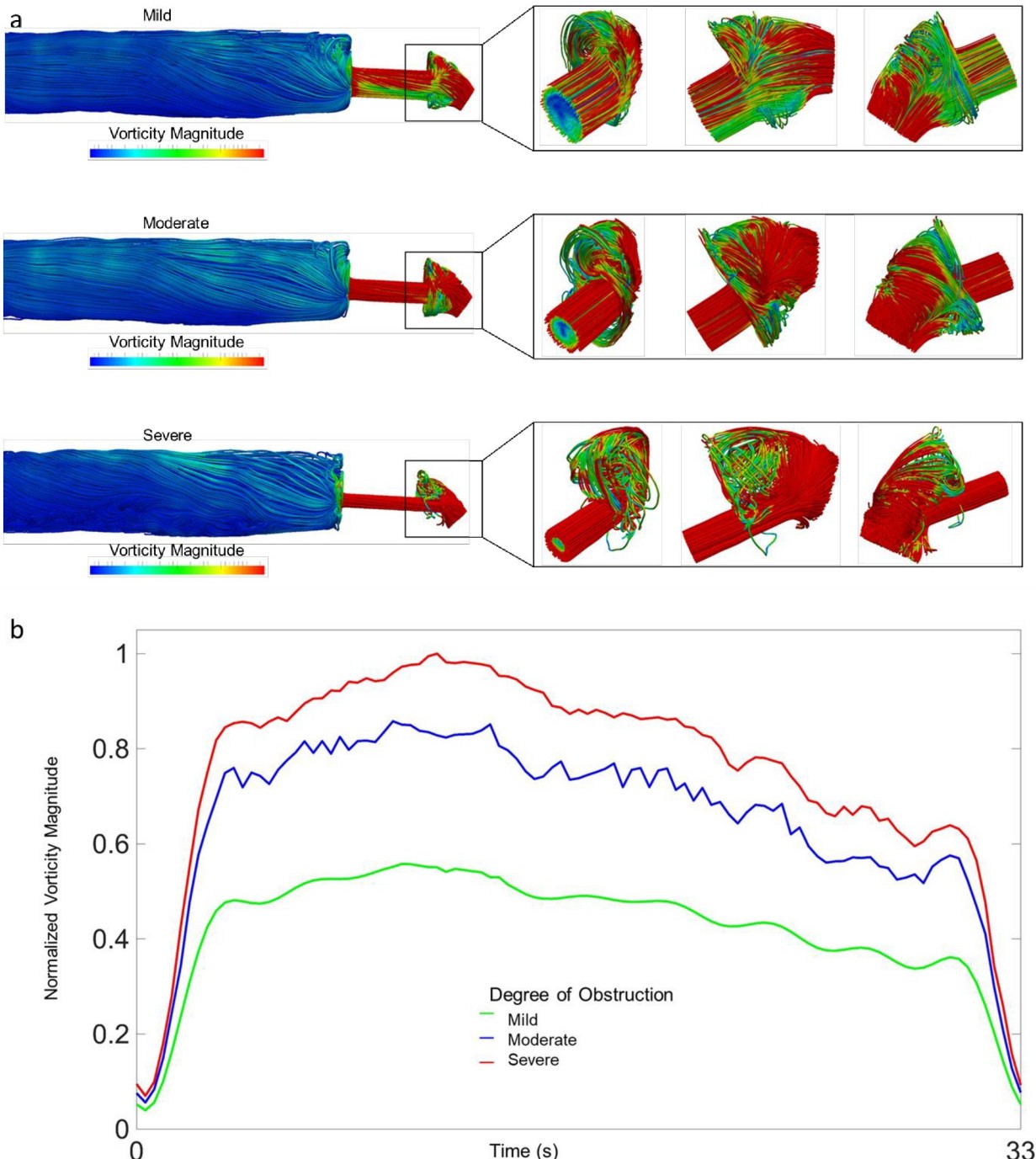

**Fig 4.** a: Computational modeling of vorticity with increasing levels of distal urethral obstruction; b: Normalized vorticity over voiding time stratified by level of obstruction; 83%, 90% and 96% reduction in lumen area for mild, moderate, and severe, respectively.

Within urology, 3D printed models have previously been used for simulation, education and surgical planning. 3D models have been particularly useful for surgical training such as practicing urethrovesical anastomosis and partial nephrectomy [15–17]. Moreover, 3D printing may become an adjunct for prostate biopsy planning [18]. Attempts to perform dynamic testing on models of human tissues may represent a new avenue of research for multiple

disciplines in which human tissue research is too expensive, unethical, or technically difficult. Given the inherent qualities of 3D printing, unlimited samples can be created. Our experiments used a rigid plastic for urethral models, but next generation flexible materials will better simulate human tissue [6,19].

The discrepancy in flow rates between cadaveric and 3D printed tissues likely was in part due to their material properties. The polymer used in this experiment had no elasticity, and essentially was a rigid pipe. The degree to which tissue elasticity impacts flow is unknown but certainly mechanical properties suggest distal urethral obstruction could be directly related to lack of tissue distensibility at high fluid flow rates [6]. The 3D printed model had no distensibility, which may explain the consistently worse flow rates observed. Of note, the cadaveric tissue, while certainly pliable, likely did not reproduce live human urethral elasticity either. Means to measure human urethral elasticity may be possible via elastography; early work in this area reveals heterogeneous results [20]. Furthermore, urethral elasticity likely is not uniform across urethral length [21]. Even if measured reliably, the limitations of 3D printing may require incorporation of dynamic elasticity be reserved for computational fluid modeling.

Linear displacement of the focal point of urine was experimentally created in our models with each fixed in position and height. We recognize that men have the ability to aim and alter urine trajectory dynamically but this was not accounted. Nonetheless, we speculate a high linear displacement may represent a stream that is difficult to direct appropriately. We observed high linear displacement for ventral urethral narrowings and low linear displacement for meatoplasty. This follows anecdotal observations from patients with meatal stenosis, but only a well-designed clinical experiment would validate these findings.

Computational fluid modeling has been utilized with great effect within cardiovascular medicine [22]. Our group applied this technique to model the idealized male urethra and urine flow [13]. This study represents the natural evolution of that work. We observed distal fluid vorticity and flow disturbance increases with worsening obstruction. These vortical motions may in part explain the experimentally observed larger spray area and linear displacement with distal obstructions. Future experiments are required to understand the impact of high pressure, high vortical flow on stricture development and progression. Future work should also investigate the relationship between repeated micro-trauma caused by high pressure voiding, subsequent squamous metaplasia and scar formation in the urethra proximal to a narrowing. Such repeated forces applied to mucosal tissue in the urethra may also produce physiologic consequence.

Ultimately our findings represent only the first iteration of a new tool to describe urine spray and spread. Uroflowmetry was used to standardize our test scenarios. Uroflowmetry results may vary and are subject to error which may have affected our results, but likely in non-differential ways [23]. Secondary spray that resulted from the initial hit of urine onto our detector bouncing onto a second location was greatly reduced but not eliminated by using moistened cloth. As such, spray areas herein may be an over-estimation. Given we report absolute and normalized spray area, this error, should be reduced and not fundamentally impact our results. We did not account for psychological effects, benign prostatic hyperplasia, varied patient heights, or bladder dysfunction in terms of the experimental production of the urinary stream. Despite our model's limitations, we have developed a reliable system to measure urinary spray and evaluate the effect of common pathologies in the distal urethra and their corrective surgeries on the urine stream. We have further characterized flow patterns in the distal urethra with fluid dynamic modeling to further explore the role of urethral stricture in spraying. Future studies would involve live human subjects to validate our findings.

## Conclusions

We present the most comprehensive study, to date, of understanding urinary spray. Distal urethral obstruction worsens flow rate and increases urine spray diameter. 3D printed models underperform relative to cadaveric tissues, nonetheless, potentially offer a cheap, safe, and reproducible manner to study the impact of surgical changes in the urethra on urinary flow. Coupled with dynamic modeling, a new understanding of urine flow is possible.

## Supporting information

**S1 Appendix.**
(TIF)

**S1 File.**
(WMV)

## Author Contributions

**Conceptualization:** Andrew J. Cohen, German Patino.

**Data curation:** German Patino, Sudarshan Srirangapatanam, Anas Tresh, Anthony Enriquez.

**Formal analysis:** Andrew J. Cohen, German Patino, Sudarshan Srirangapatanam.

**Investigation:** Andrew J. Cohen, Anas Tresh, Bhagat Cheema, Anthony Enriquez.

**Methodology:** Andrew J. Cohen, German Patino, Mehran Mirramezani, Anas Tresh, Bhagat Cheema, Anthony Enriquez.

**Project administration:** Anthony Enriquez.

**Resources:** Andrew J. Cohen, Mehran Mirramezani, Laurence S. Baskin, Shawn C. Shadden.

**Supervision:** Benjamin N. Breyer.

**Visualization:** Mehran Mirramezani, Jenny Tai, Dylan Romero, Shawn C. Shadden.

**Writing – original draft:** Andrew J. Cohen.

**Writing – review & editing:** Andrew J. Cohen, German Patino, Sudarshan Srirangapatanam, Shawn C. Shadden.

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
