## [Decision Letter · Decision Letter 0]

28 Jul 2020

PONE-D-20-16734

Novel measurement tool and model for aberrant urinary stream in 3D printed urethras derived from human tissue

PLOS ONE

Dear Dr. Enriquez,

Thank you for submitting your manuscript to PLOS ONE. After careful consideration, we feel that it has merit but does not fully meet PLOS ONE’s publication criteria as it currently stands. Therefore, we invite you to submit a revised version of the manuscript that addresses the points raised during the review process.

ACADEMIC EDITOR:

In addition to the reviewers comments I ask you to consider to rephrase "altered the shape of the urethra' in the patient summary. Furthermore: because meatal stenosis (shape -size variation) is very much different from pelvic muscle and or prostatic outflow obstruction a short discussion about flow -rate (which uroflowmetry measures) and or flow velocity (which the 'nozzle' (meatus) affects) might help the reader to understand the fundamental limitations of urodlowmetry for -diagnosis of- meatal abnormalities.

We look forward to receiving your revised manuscript.

Kind regards,

Peter F.W.M. Rosier, M.D. PhD

Academic Editor

PLOS ONE

Journal Requirements:

Additional Editor Comments (if provided):

Reviewers' comments:

Reviewer's Responses to Questions

**Comments to the Author**

1. Is the manuscript technically sound, and do the data support the conclusions?

Reviewer #1: Yes

Reviewer #2: Yes

2. Has the statistical analysis been performed appropriately and rigorously? 

Reviewer #1: Yes

Reviewer #2: Yes

3. Have the authors made all data underlying the findings in their manuscript fully available?

Reviewer #1: Yes

Reviewer #2: Yes

4. Is the manuscript presented in an intelligible fashion and written in standard English?

Reviewer #1: Yes

Reviewer #2: Yes

5. Review Comments to the Author

Reviewer #1: This paper is well written, easy to understand and thorough in its description and discussion.

The only significant matters requiring attention are I think:

- p.3 Intro 1st para: you claim there is no instrument, but ref 6 does describe something of this sort. You also neglect to mention the drop spectrometer (doi: 10.1109/tbme.1976.324642), an acknowledgement of which might be appropriate.

- The difference in flow rates between cadaver and model is clearly significant. You should in this context discuss the difference in elasticity between tissue and polymer, which is likely to be a cause. Otherwise, this difference will seem to invalidate the model

- the outlet of the model will clearly be a major factor for spraying. I think some reference to any rounding / shaping (or not) of this that you did will help understand.

- the CFD modeling does not really fit with this paper, and would easily make a separate publication if you chose to do so.

Other smaller errors to attend to:

- unit of flow is correctly ml/s in Table 1, but is not consistent through the paper

- ref 10 on p4 seems to be too late in the sentence

- p5 references to equipment should normally include manufacturer and location

- p6: 'rectangular' should be 'rectangle'?

- p11, 1st para: 'recapitulated' = ?

- p12 2nd para: 'vertical' = 'vortical'?

- Fig 2a: 'calibration over catheter' = ?

- Fig 3: 'Alternations' 'sprauy'

Reviewer #2: Summary: The authors forced a certain flow rate (4 discrete values) through 4 human cadaveric urethral specimens and 3D printed models of these. The tip of the urethras was altered in 8 different ways (degrees of obstruction). The effects on the flow rate and the spray of the urine was studied as well as the similarity between the specimens and the 3D models. Finally, computational fluid dynamics was applied to study intra-urethral pressure and fluid vorticity. The authors describe 5 objectives of their study. The main aim was to provide models for studying the best surgical approach to correct anatomical abnormalities of the meatus.

Major comments:

1. It is doubtful whether the cadaveric urethra is a good model for the living urethra because of the absence of muscle tone. This is not discussed. In addition, there appear to be large differences in Qmax values and spray areas between the cadaveric urethras and the 3D models (Table 1), raising the same question.

2. I found it sometimes difficult to follow what the authors exactly did and I suppose this is partly due to the presentation of data. Why not, for instance, is Appendix 2 incorporated in Table 1. Also, presentation in a graph might be more explaining than presentation in a table.

3. In spite of 1 and 2, I can agree with the authors that 3D models could be used to study the clinical effect of different surgical approaches of a given urethral abnormality before application in patients.

Minor comments:

1. Abstract, Results and Limitations, last sentence: 2x was.

2. There is a discrepancy between the text in Results and Table 1: 3D printed models had …: 10.6 ml/sec should be 11.6 ml/sec (Table 1).

6. PLOS authors have the option to publish the peer review history of their article (what does this mean?). If published, this will include your full peer review and any attached files.

Reviewer #1: Yes: Andrew Gammie

Reviewer #2: Yes: Jan Groen

---

## [Author Response · Author response to Decision Letter 0]

27 Aug 2020

5. Review Comments to the Author

Reviewer #1: This paper is well written, easy to understand and thorough in its description and discussion.

The only significant matters requiring attention are I think:

- p.3 Intro 1st para: you claim there is no instrument, but ref 6 does describe something of this sort. You also neglect to mention the drop spectrometer (doi: 10.1109/tbme.1976.324642), an acknowledgement of which might be appropriate.

Thank you for your comments and suggestion. We appreciate the historic, research significance of the drop spectrometer. We have added references for this series of work in multiple points throughout the present work. 

- The difference in flow rates between cadaver and model is clearly significant. You should in this context discuss the difference in elasticity between tissue and polymer, which is likely to be a cause. Otherwise, this difference will seem to invalidate the model

Thank you. You astutely identify the major limitation of this work. The polymer used in this experiment had no elasticity, and essentially was a rigid pipe. The cadaveric tissue, while pliable, likely does not recapitulate live human urethral elasticity given lack of muscle tone. We hypothesis this is why the flow rates between cadaver and models were significantly different. 

We experimentally determined the cadaveric urethral mucosa deformed by 2-3 mm during dynamic testing by observation with ultrasound. The material for the 3D models, in contrast, was non-pliable. Moreover, for simplicity the CFD model simulation assumed a non-pliable urethra. The elasticity of normal human urethral tissue at the tip of the penis is unknown. Histologic analysis confirms the presence of elastic fibers. Detailed biomechanical analysis of animals, such as horse urethra may or may not be translatable to human models. Therefore, incorporation of elasticity in any model remains a challenge, partly due to lack of basic knowledge in this arena. Means to measure human urethral elasticity may be possible via elastography. This has been performed on the prostatic urethra with quite heterogeneous results. Given the lack of gold-standard references, it is hard to incorporate such data here. While complex deformable polymers are now possible with 3D printing, cost limitations remain a concern. Furthermore, urethral elasticity likely is not uniform across the urethral length and how to incorporate dynamic elasticity to fully model human tissue may only be possible via simulation. These ideas are summarized for the reader on page 12 in a new paragraph. 

“The discrepancy in flow rates between cadaveric and 3D printed tissues likely was in part due to their material properties. The polymer used in this experiment had no elasticity, and essentially was a rigid pipe. The degree to which tissue elasticity impacts flow is unknown but certainly mechanical properties suggest distal urethral obstruction could be directly related to lack of tissue dispensability at high fluid flow rates.(6) The 3D printed model had no dispensability, which may explain the consistently worse flow rates observed. Of note, the cadaveric tissue, while certainly pliable, likely did not recapitulate live human urethral elasticity either. Means to measure human urethral elasticity may be possible via elastography; early work in this area reveals heterogeneous results.(20) Furthermore, urethral elasticity likely is not uniform across urethral length. (21) Even if measured reliably, the limitations of 3D printing may require incorporation of dynamic elasticity be reserved for computational fluid modeling.”

Outside of the scope of the article but of interest to the reviewer may be the ongoing research in measuring ‘stiffness’ in arteries. Shear wave elastography has been proposed as a method to study arterial wall elasticity. To our knowledge this has never been applied to the human urinary tract, but could be a modality to gain enough information to incorporate in the future. 

Bastos AL, Silva EA, Silva Costa W, et al. The concentration of elastic fibres in the male urethra during human fetal development. BJU International 2004;94:620–623. doi:10.1111/j.1464-410X.2004.05012.x.

Messas E, Pernot M, Couade M. Arterial wall elasticity: State of the art and future prospects. Diagnostic and Interventional Imaging 2013;94:561–569. doi:10.1016/j.diii.2013.01.025.

Natali AN, Carniel EL, Frigo A, et al. Experimental investigation of the biomechanics of urethral tissues and structures: Biomechanics of urethral tissues and structures. Exp Physiol 2016;101:641–656. doi:10.1113/EP085476.

Kwon JK, Kim DK, Lee JY, et al. Relationship between Lower Urinary Tract Symptoms and Prostatic Urethral Stiffness Using Strain Elastography: Initial Experiences. JCM 2019;8:1929. doi:10.3390/jcm8111929.

- the outlet of the model will clearly be a major factor for spraying. I think some reference to any rounding / shaping (or not) of this that you did will help understand.

We have edited the methods section to further aid in understanding of the varied shape confirmations that we enacted. Furthermore the Fig x has been granted additional explanation to improve the readability and understanding of the methods. Ultimately, the alterations were based on mucosa to mucosa apposition. Hence the original organic shape of the meatus would remain, simply with smaller aperture. 

- the CFD modeling does not really fit with this paper, and would easily make a separate publication if you chose to do so.

 We appreciate that the CFD modeling is a complex topic potentially worthy of its own focused manuscript. In constructing the 3D printed models, the 3D representation of the urethral lumen was created. Moreover, the CFD models recapitulate the findings of 3D printed urethral experiments and offer tantalizing clues to the internal flow within the lumen. In presenting this compendium of methodologies together, we feel the reader can best critically appraise how these inter-connected methods may lead to broadened clinical understanding and application. 

Other smaller errors to attend to:

- unit of flow is correctly ml/s in Table 1, but is not consistent through the paper

Thank you. We have corrected such unit omissions through the manuscript for consistency and scientific vigor. 

- ref 10 on p4 seems to be too late in the sentence

This has been corrected 

- p5 references to equipment should normally include manufacturer and location

This oversight has been corrected

- p6: 'rectangular' should be 'rectangle'? Corrected

- p11, 1st para: 'recapitulated' = ? Corrected

- p12 2nd para: 'vertical' = 'vortical'? Corrected 

- Fig 2a: 'calibration over catheter' = ? This has been further explained in the methods and the fig legend. 

- Fig 3: 'Alternations' 'sprauy' This has been corrected 

Reviewer #2: Summary: The authors forced a certain flow rate (4 discrete values) through 4 human cadaveric urethral specimens and 3D printed models of these. The tip of the urethras was altered in 8 different ways (degrees of obstruction). The effects on the flow rate and the spray of the urine was studied as well as the similarity between the specimens and the 3D models. Finally, computational fluid dynamics was applied to study intra-urethral pressure and fluid vorticity. The authors describe 5 objectives of their study. The main aim was to provide models for studying the best surgical approach to correct anatomical abnormalities of the meatus.

We very much appreciate you reading the manuscript and found your comments extremely helpful. 

Major comments:

1. It is doubtful whether the cadaveric urethra is a good model for the living urethra because of the absence of muscle tone. This is not discussed. In addition, there appear to be large differences in Qmax values and spray areas between the cadaveric urethras and the 3D models (Table 1), raising the same question.

We appreciate this concern; together with the reviewer’s 1’s similar reservations about the elasticity of the model we have added a discussion paragraph (page 12) to directly address this issue. We agree a major limitation is the fact the polymer used in this experiment had no elasticity, and essentially was a rigid pipe. The cadaveric tissue likely does not fully recapitulate live human urethral elasticity. 

2. I found it sometimes difficult to follow what the authors exactly did and I suppose this is partly due to the presentation of data. Why not, for instance, is Appendix 2 incorporated in Table 1. Also, presentation in a graph might be more explaining than presentation in a table.

We tested multiple presentation of data strategies with a wide audience of experts, clinicians, and scientists given the complexity of the tip alterations and the data generated. After circulating several draft graphics and tables it was the consensus opinion to use the current hybrid approach. Hence, we have broken up the data into Table 1 and Figure 3 to illustrate our points. 

Nonetheless, we are very open to merging Table 1 and Appendix 2 to aid in understanding. As such a new Table 1 has been uploaded and appendix 2 eliminated. 

3. In spite of 1 and 2, I can agree with the authors that 3D models could be used to study the clinical effect of different surgical approaches of a given urethral abnormality before application in patients.

Minor comments:

1. Abstract, Results and Limitations, last sentence: 2x was.

Thank you for pointing out this error, it his has been corrected

2. There is a discrepancy between the text in Results and Table 1: 3D printed models had …: 10.6 ml/sec should be 11.6 ml/sec (Table 1).

We appreciate you finding this mis-type and it has been corrected.

---

## [Decision Letter · Decision Letter 1]

28 Sep 2020

PONE-D-20-16734R1

Novel measurement tool and model for aberrant urinary stream in 3D printed urethras derived from human tissue

PLOS ONE

Dear Dr. Enriquez,

Thank you for submitting your manuscript to PLOS ONE. After careful consideration, we feel that it has merit but does not fully meet PLOS ONE’s publication criteria as it currently stands. Therefore, we invite you to submit a revised version of the manuscript that addresses the points raised during the review process.

ACADEMIC EDITOR:

The reviewer suggests a fes changes in wording. I agree with these, can you adapt the manuscript?

We look forward to receiving your revised manuscript.

Kind regards,

Peter F.W.M. Rosier, M.D. PhD

Academic Editor

PLOS ONE

Reviewers' comments:

Reviewer's Responses to Questions

**Comments to the Author**

1. If the authors have adequately addressed your comments raised in a previous round of review and you feel that this manuscript is now acceptable for publication, you may indicate that here to bypass the “Comments to the Author” section, enter your conflict of interest statement in the “Confidential to Editor” section, and submit your "Accept" recommendation.

Reviewer #1: All comments have been addressed

Reviewer #2: (No Response)

2. Is the manuscript technically sound, and do the data support the conclusions?

Reviewer #1: Yes

Reviewer #2: Yes

3. Has the statistical analysis been performed appropriately and rigorously? 

Reviewer #1: Yes

Reviewer #2: Yes

4. Have the authors made all data underlying the findings in their manuscript fully available?

Reviewer #1: Yes

Reviewer #2: Yes

5. Is the manuscript presented in an intelligible fashion and written in standard English?

Reviewer #1: Yes

Reviewer #2: Yes

6. Review Comments to the Author

Reviewer #1: Thank you for carefully considering and addressing the comments made. There are just some small remaining issues to sort:

You use 'dispensability' twice now on p.12, whereas I think you mean 'distensibility'.

In the new text on p.12, you use 'recapitulate' which may be a valid technical term in modelling, but it does not communicate well here. I suggest 'did not reproduce live human tissue' is clearer, as you have done in the abstract.

Your comment on shear wave elastography is interesting, and you may be interested to check out J Urol 2017; 198:422-429. I agree that this method may well be worth future investigation.

In Tables 1 and 2, the addition of horizontal and vertical lines may assist understanding. I found it not immediately obvious which figures the p-values corresponded with, and also where the division between specimens lay.

Reviewer #2: Discussion, first paragraph: intrauthreal should be corrected.

In the Discussion, more attention is paid now to the differences between living tissue, cadaveric urethras and 3D models, that is, to the limitations of the study. There are large differences between results obtained in the cadaveric urethras and the 3D models. Nevertheless, this study seems a good onset of a series of experiments and observations on the value of 3D models in clinical practice.

7. PLOS authors have the option to publish the peer review history of their article (what does this mean?). If published, this will include your full peer review and any attached files.

Reviewer #1: **Yes: **Andrew Gammie

Reviewer #2: **Yes: **Jan Groen

---

## [Author Response · Author response to Decision Letter 1]

6 Oct 2020

Reviewer #1: Thank you for carefully considering and addressing the comments made. There are just some small remaining issues to sort:

You use 'dispensability' twice now on p.12, whereas I think you mean 'distensibility'.

In the new text on p.12, you use 'recapitulate' which may be a valid technical term in modelling, but it does not communicate well here. I suggest 'did not reproduce live human tissue' is clearer, as you have done in the abstract.

Thank you again for your helpful comments and keen eye for errors. We have made these corrections on page 12 .

Your comment on shear wave elastography is interesting, and you may be interested to check out J Urol 2017; 198:422-429. I agree that this method may well be worth future investigation.

In Tables 1 and 2, the addition of horizontal and vertical lines may assist understanding. I found it not immediately obvious which figures the p-values corresponded with, and also where the division between specimens lay.

We have added some vertical and horizontal lines to aid in drawing the readers eye. We are open to any and all table formatting suggestions to bring the paper in line with PLOSone formatting standards without reservation. 

Reviewer #2: Discussion, first paragraph: intrauthreal should be corrected.

Thank you, this spelling error corrected. 

In the Discussion, more attention is paid now to the differences between living tissue, cadaveric urethras and 3D models, that is, to the limitations of the study. There are large differences between results obtained in the cadaveric urethras and the 3D models. Nevertheless, this study seems a good onset of a series of experiments and observations on the value of 3D models in clinical practice.

---

## [Decision Letter · Decision Letter 2]

16 Oct 2020

Novel measurement tool and model for aberrant urinary stream in 3D printed urethras derived from human tissue

PONE-D-20-16734R2

Dear Dr. Enriquez,

We’re pleased to inform you that your manuscript has been judged scientifically suitable for publication and will be formally accepted for publication once it meets all outstanding technical requirements.

Kind regards,

Peter F.W.M. Rosier, M.D. PhD

Academic Editor

PLOS ONE

Additional Editor Comments (optional):

Reviewers' comments:

Reviewer's Responses to Questions

**Comments to the Author**

1. If the authors have adequately addressed your comments raised in a previous round of review and you feel that this manuscript is now acceptable for publication, you may indicate that here to bypass the “Comments to the Author” section, enter your conflict of interest statement in the “Confidential to Editor” section, and submit your "Accept" recommendation.

Reviewer #1: All comments have been addressed

2. Is the manuscript technically sound, and do the data support the conclusions?

Reviewer #1: (No Response)

3. Has the statistical analysis been performed appropriately and rigorously? 

Reviewer #1: (No Response)

4. Have the authors made all data underlying the findings in their manuscript fully available?

Reviewer #1: (No Response)

5. Is the manuscript presented in an intelligible fashion and written in standard English?

Reviewer #1: (No Response)

6. Review Comments to the Author

Reviewer #1: (No Response)

7. PLOS authors have the option to publish the peer review history of their article (what does this mean?). If published, this will include your full peer review and any attached files.

Reviewer #1: **Yes: **Andrew Gammie

---

## [Editor Report · Acceptance letter]

30 Oct 2020

PONE-D-20-16734R2 

Novel Measurement Tool and Model for Aberrant Urinary Stream in 3D Printed Urethras derived from Human Tissue 

Dear Dr. Enriquez:

I'm pleased to inform you that your manuscript has been deemed suitable for publication in PLOS ONE. Congratulations! Your manuscript is now with our production department. 

Kind regards, 

on behalf of

Dr. Peter F.W.M. Rosier 

Academic Editor

PLOS ONE